# The zinc finger proteins ZNF644 and WIZ regulate the G9a/GLP complex for gene repression

**Chunjing Bian†, Qiang Chen†, Xiaochun Yu***

Division of Molecular Medicine and Genetics, Department of Internal Medicine, University of Michigan Medical School, Ann Arbor, United States

**Abstract** The G9a/GLP complex mediates mono- and dimethylation of Lys9 of histone H3 at specific gene loci, which is associated with transcriptional repression. However, the molecular mechanism by which the G9a/GLP complex is targeted to the specific gene loci for H3K9 methylation is unclear. In this study, with unbiased protein affinity purification, we found ZNF644 and WIZ as two core subunits in the G9a/GLP complex. ZNF644 and WIZ interact with the transcription activation domain of G9a and GLP, respectively. Moreover, both ZNF644 and WIZ contain multiple zinc finger motifs that recognize consensus DNA sequences. ZNF644 and WIZ target G9a and GLP to the chromatin and mediate the G9a/GLP complex-dependent H3K9 methylation as well as gene repression. Thus, our studies reveal two key subunits in the G9a/GLP complex that regulate the function of this histone methyltransferase complex.

**\*For correspondence:** xiayu@umich.edu

†These authors contributed equally to this work

**Competing interests:** The authors declare that no competing interests exist.

## Introduction

Post-translational modifications of histones, especially lysine methylation, play important roles in diverse biological processes, such as gene transcription, chromatin packaging, and cellular differentiation. Different lysine residues of histones are methylated by different histone lysine-specific methyltransferases (HKMTs). Among HKMTs, G9 and GLP specifically catalyze H3K9 mono- (H3K9me1) and dimethylation (H3K9me2) (*Tachibana et al., 2001*, *2005*; *Kubicek et al., 2007*).

G9a and GLP are paralogs with similar domain architecture. Both contain a transcription activation domain (TAD), a glutamate-rich domain, a cysteine-rich domain, 7 tandem ankyrin repeats (ANKs), and a methyltransferase domain (*Roopra et al., 2004*; *Dillon et al., 2005*; *Lee et al., 2006*; *Purcell et al., 2011*; *Shinkai and Tachibana, 2011*; *Bittencourt et al., 2012*). Interestingly, G9a and GLP form a heterodimer via the interaction between the C-terminal catalytic domains (*Tachibana et al., 2005*). The major function of the G9a/GLP complex is to catalyze H3K9me1 and H3K9me2 in euchromatin, which is associated with transcriptional repression (*Tachibana et al., 2005*, *2008*). Accumulated evidence has shown that this methyltransferase complex regulates multiple biological processes, such as meiosis, embryonic development, immune response, and tumorigenesis (*Tachibana et al., 2002*, *2007*; *Schaefer et al., 2009*; *Chen et al., 2010*; *Huang et al., 2010*; *Shinkai and Tachibana, 2011*). Interestingly, if one of these two methyltransferases is deleted, the other one alone has little enzymatic activity in vivo, suggesting that the heterodimer formation is important for the function of this enzyme complex (*Tachibana et al., 2005*, *2008*). Downstream functional partners of the G9a/GLP complex have been examined. Since HP1 recognizes H3K9me2 (*Bannister et al., 2001*; *Lachner et al., 2001*; *Nielsen et al., 2001*), it is likely that the G9a/GLP complex mediates the recruitment of HP1 to specific gene loci for transcriptional repression (*Ogawa et al., 2002*; *Nishio and Walsh, 2004*; *Shinkai and Tachibana, 2011*). Moreover, the ANKs of G9a and GLP also recognize H3K9me1 and H3K9me2 (*Collins et al., 2008*), which may facilitate the chromatin spreading of the G9a/GLP complex.

**eLife digest** Genes encode instructions for processes within cells, but only a small subset of the genes within a cell will be switched on (or expressed) at any given time. The other genes are kept switched off until their instructions are needed. For example, some genes are switched on when it is time for a cell to divide or in response to changes in the environment.

In humans and other eukaryotes, DNA is packaged within cells in proteins called histones. The level of gene expression can be altered by how tightly the DNA is packaged; if the DNA is more tightly packed around the histones, the gene will be expressed at lower levels than if the DNA is only loosely packed.

A group of proteins called the G9a/GLP complex can alter histones to reduce the expression of some genes during embryo development, immune responses, and the formation of tumors. The complex works by attaching 'methyl' tags to the histones associated with particular genes, but it is not clear how it is able to specifically target these histones.

Bian, Chen, and Yu used a technique called unbiased protein affinity purification to search for other proteins that can bind to the G9a/GLP complex. The experiments found two proteins called ZNF644 and WIZ, both of which are required for the G9a/GLP complex to be able to add methyl tags to histones.

Further experiments revealed that ZNF644 and WIZ both contain regions called zinc finger motifs that enable them to identify and bind to specific sequences of DNA. Therefore, these proteins can guide the G9a/GLP complex to specific sites in the genome to switch off the expression of particular genes. A future challenge will be to try to modify these zinc finger motifs and guide the G9a/GLP complex to switch off other genes. This may allow us to develop therapies that could alter the expression of genes involved in cancer and other diseases.

Recent study suggests that the G9a/GLP complex is associated with Polycomb Repressive Complex 2 (PRC2) (*Mozzetta et al., 2014*). EZH2, the catalytic subunit in the PRC2, regulates histone H3K27 methylation. Thus, it is likely that these methyltransferases function together to modulate histone codes during transcription. In addition to histone methylation, the G9a/GLP complex also regulates DNA methylation during early embryogenesis, which is independent of their methyl-transferase activity. It has been reported that the G9a/GLP complex associates with DNMT1 through PCNA (*Esteve et al., 2006*). Moreover, the ANKs of G9a interact with DNMT3A and 3B for the de novo DNA methylation (*Epsztejn-Litman et al., 2008*; *Chang et al., 2011*).

Although the G9a/GLP complex plays an important role in epigenetic modification and gene transcription, the molecular mechanism by which the G9a/GLP complex is regulated in vivo remains elusive. Although several partners of G9a have been identified, it is unclear whether these partners form a stable complex with G9a and GLP, and directly control the G9a/GLP-dependent H3K9me1 and H3K9me2. In this study, we searched other possible subunit(s) in the G9a/GLP complex. With unbiased protein affinity purification, we found that ZNF644 and WIZ, two zinc finger proteins, interact with G9a and GLP, respectively. These two zinc finger proteins target G9a and GLP to genomic loci for the regulation of gene transcription.

## Results

### ZNF644 and WIZ are binding partners of G9a

To explore the regulation mechanism of the G9a/GLP complex, we have searched the functional partner(s) of G9a using tandem protein affinity purification. Cell lysates of 293T cells stably expressing SFB-tagged G9a were subjected to two rounds of affinity purification. Since G9a and GLP form a heterodimer in vivo (*Tachibana et al., 2005*), we could easily detect GLP as a partner of G9a in this purification, which was served as a positive control. Interestingly, besides GLP, G9a also interacted with two other proteins. Mass spectrometry analysis revealed that these two proteins were ZNF644 and WIZ, two zinc finger proteins. Between these two proteins, WIZ has been known to regulate the stability of the G9a/GLP heterodimer (*Ueda et al., 2006*), while the function of ZNF644 has not been characterized yet (*Figure 1A*). To validate our initial purification results, we performed reciprocal

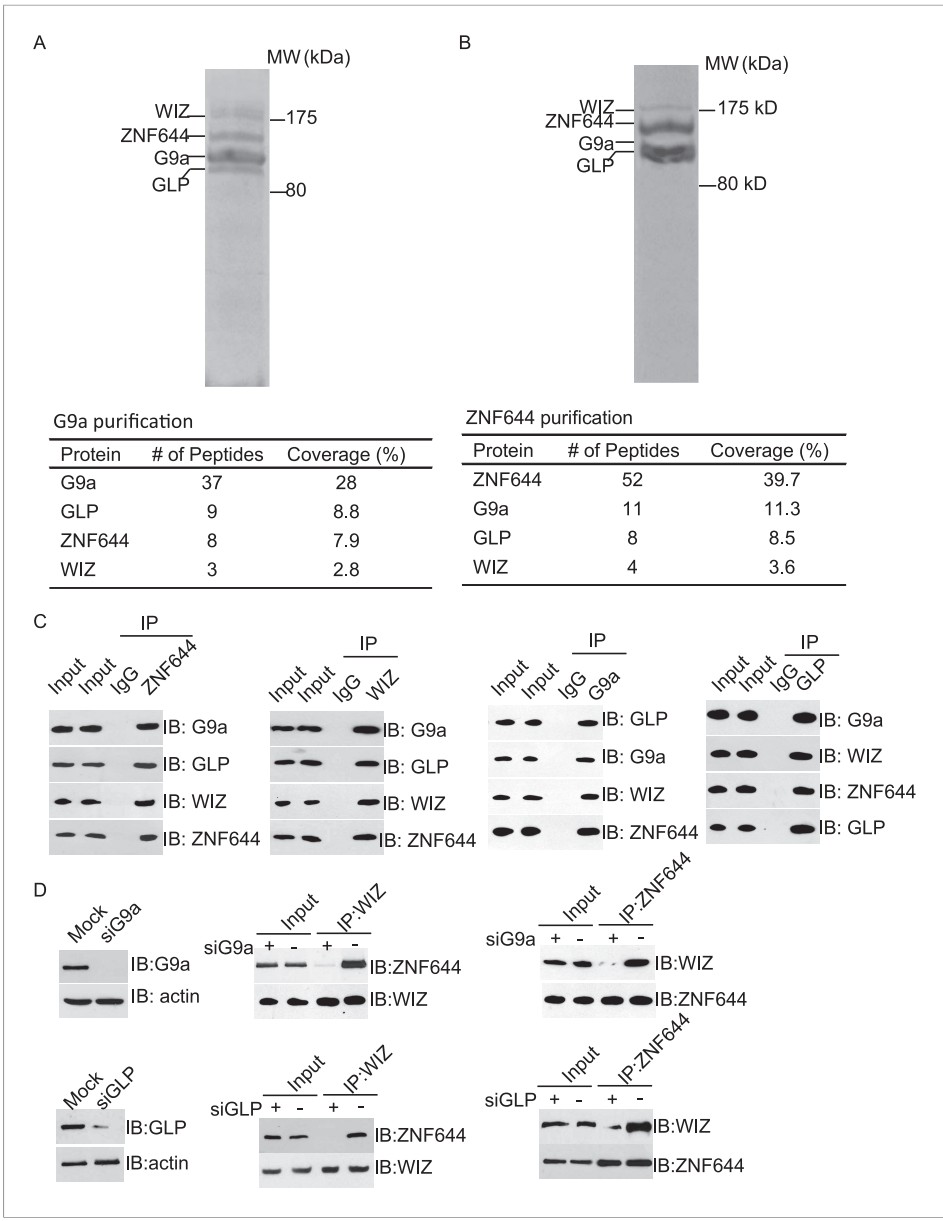

**Figure 1**. ZNF644 and WIZ associate with G9a. (**A**) Silver staining of affinity-purified G9a complex. Cell lysates of 293T cells stably expressing SFB-G9a were subjected to affinity purification. Eluted proteins were visualized by silver staining. Arrows indicate proteins corresponding to G9a, GLP, ZNF644 and WIZ. Peptide coverage is shown in the table. (**B**) Silver staining of affinity-purified ZNF644 partners. (**C**) ZNF644 and WIZ co-exist in the same complex with G9a and GLP. U2OS cell lysates were analyzed by co-immunoprecipitation (co-IP) and Western blotting with the antibodies indicated. The whole cell lysates of U2OS was used as the input. An irrelevant IgG was used as the IP control. (**D**) Down-regulation of G9a or GLP impairs the interaction between WIZ and ZNF644. G9a and GLP were down-regulated by siRNAs in U2OS cells. The cell lysates were analyzed by IP and Western blotting with the antibodies indicated.

The following figure supplement is available for figure 1:

**Figure supplement 1**. ZNF644 and WIZ antibodies have been generated and specifically recognize the endogenous ZNF644 and WIZ respectively.

affinity purification using ZNF644 as the bait. Again, we identified G9a, GLP, and WIZ as binding partners of ZNF644, suggesting that ZNF644 and WIZ co-exist in the same complex with G9a and GLP (*Figure 1B*).

To further confirm the interaction between these proteins in vivo, we first raised antibodies against endogenous ZNF644 or WIZ. Anti-ZNF644 and WIZ antibodies specifically recognized bands around 150 kDa and 175 kDa, respectively. Moreover, siZNF644 and siWIZ treatment diminished the expression of these two proteins, indicating that both antibodies specifically recognize the endogenous proteins (*Figure 1—figure supplement 1*). We next performed co-immunoprecipitation (co-IP) assays using U2OS cell lysates and found that one protein associated with the other proteins in this complex (*Figure 1C*), suggesting that these proteins are core subunits in the G9a/GLP complex. To further characterize the interactions between these subunits, we knocked down G9a or GLP by siRNAs. Interestingly, down-regulation of G9a or GLP impaired the interaction between WIZ and ZNF644 (*Figure 1D*), suggesting that the association between WIZ and ZNF644 is mediated by G9a and GLP. Collectively, by unbiased protein affinity purification and co-IP assays, we found that ZNF644 and WIZ are two important subunits in the G9a/GLP complex.

## Transcription activation domains of G9a and GLP interact with ZNF644 and WIZ, respectively

Next, we examined the interaction domain in each subunit in this complex. Based on the domain architecture (*Lee et al., 2006*; *Shinkai and Tachibana, 2011*), we generated four internal deletion mutants of G9a to delete the TAD, the Glu-rich and Cys-rich domains, the ANKs and the catalytic domain, respectively (*Figure 2A*). Interestingly, the D1 mutant of G9a abolished the interaction with ZNF644 (*Figure 2B*). Since the D1 mutant of G9a lacks the TAD, it suggests that ZNF644 interacts with the TAD of G9a. Moreover, lacking the catalytic domain of G9a disrupted the interaction with WIZ (*Figure 2C*). However, the catalytic domain of G9a also interacts with the catalytic domain of GLP for a heterodimer. Thus, it is possible that the association between G9a and WIZ is mediated by GLP. To confirm this hypothesis, we knocked down G9a by siRNA (*Figure 2—figure supplement 1A*). Lacking G9a, WIZ still interacted with GLP (*Figure 2—figure supplement 1B*), suggesting that WIZ is likely to directly interact with GLP. To further elucidate the interactions between these subunits, we generated two internal deletion mutants of GLP to delete either the TAD or the catalytic domain (*Figure 2D*). Only the TAD of GLP, but not the catalytic domain of GLP, is required for the interaction with WIZ (*Figure 2E*), suggesting that WIZ recognizes the TAD of GLP. In contrast, the catalytic domain of GLP is required for the interaction with ZNF644 (*Figure 2F*). Since the catalytic domains of G9a and GLP form a heterodimer, it is likely that ZNF644 directly recognizes the TAD of G9a and associates with GLP via the interactions between G9a and GLP. Taken together, ZNF644 and WIZ interact with the TADs of G9a and GLP, respectively.

We also mapped the interaction regions on WIZ and ZNF644 by generating series of internal deletion mutants of ZNF644 and WIZ, respectively (*Figure 2G,H*). The D1 mutant of ZNF644 abolished the interaction with G9a (*Figure 2G*), suggesting that the N-terminus of ZNF644 interacts with the TAD of G9a. In contrast, the C-terminus of WIZ is required for the interaction with GLP (*Figure 2H*). Taken together, with the analyses on the internal deletion mutants, we found that the N-terminus of ZNF644 interacts with the TAD of G9a, while the C-terminus of WIZ interacts with the TAD of GLP (*Figure 2I*). Since both ZNF644 and WIZ have multi zinc-finger motifs, it is likely that both ZNF644 and WIZ regulate the function of G9 and GLP.

## Deposition of G9a on the chromatin depends on ZNF644 and WIZ

Since zinc finger motif is a DNA-binding module (*Klug and Rhodes, 1987*), we ask if ZNF644 and WIZ associate with chromatin. We lysed cells with NETN100 solution (0.5% NP-40, 2 mM EDTA, 10 mM Tris–HCl pH 8.0, and 100 mM NaCl). However, both ZNF644 and WIZ could not be eluted into soluble fraction under low salt conditions (*Figure 3A*). Interestingly, after treating the insoluble pellets with Benzonase to digest the genomic DNA, both ZNF644 and WIZ were eluted into the soluble fraction, suggesting that ZNF644 and WIZ are chromatin-bound proteins. Usually, chromatin-bound proteins could be eluted from genomic DNA by 300 mM NaCl treatment (*Zhang et al., 2009*; *Chen et al., 2013*). With increased sodium concentration in the lysis buffer, ~ 50% of ZNF644 was eluted out. However, only a small fraction of WIZ could be eluted from the chromatin, and the remaining WIZ was

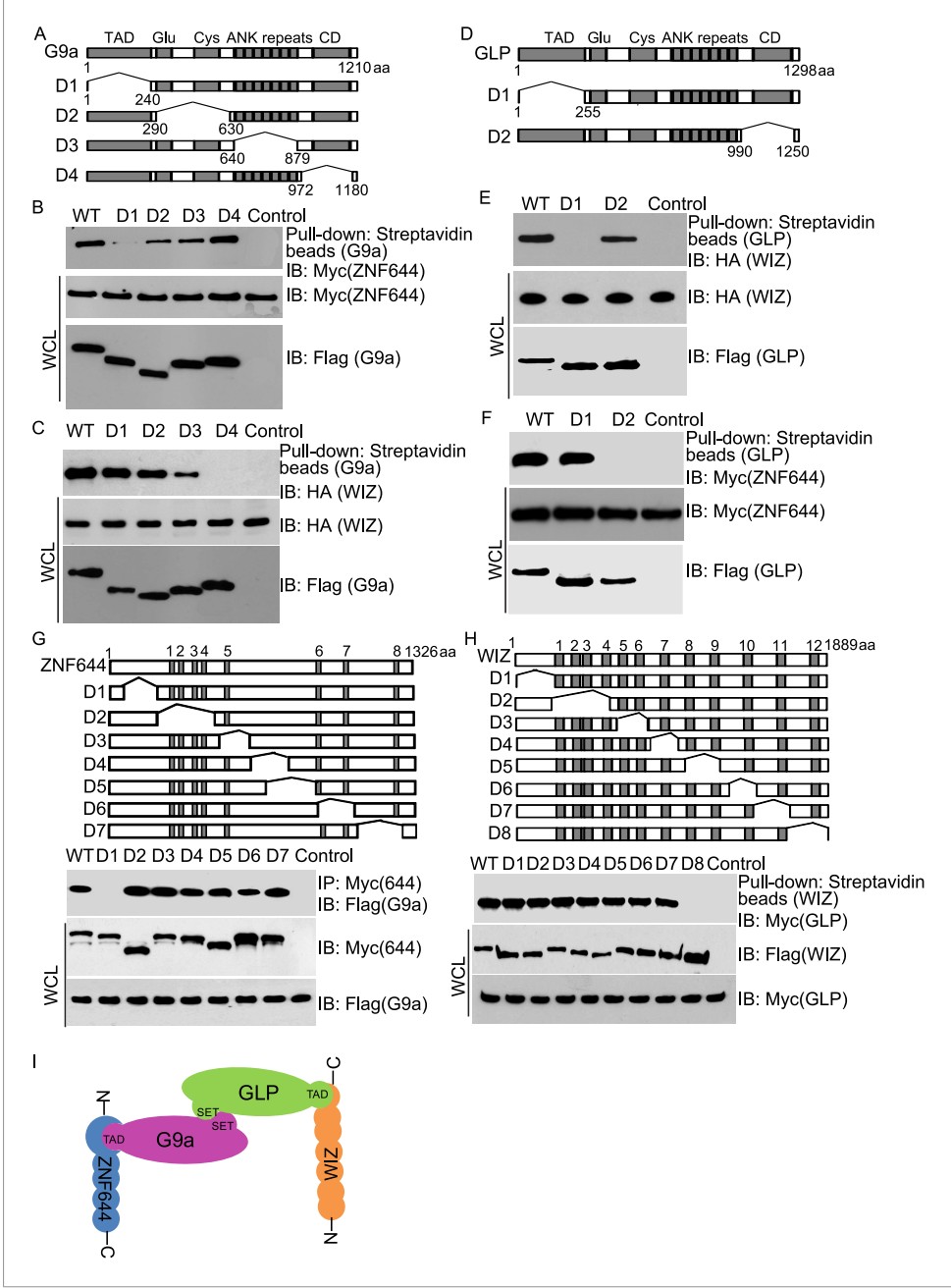

**Figure 2**. Mapping the interaction regions of ZNF644, WIZ, G9a and GLP. (**A**) A series of deletion mutants of SFB-tagged G9a were generated to map the interaction region of G9a. CD: catalytic domain. (**B**) The D1 mutant of G9a abolishes the interaction with ZNF644. SFB-tagged wild-type G9a and deletion mutants were expressed in 293T cells together with Myc-ZNF644. The cell lysates were subjected to streptavidin beads pull-down and Western blotting with the indicated antibodies. The whole cell lysates were used as the input. Cells only expressing Myc-ZNF644 were used for pull-down control. (**C**) Lacking the catalytic domain of G9a (CD) disrupts the interaction with WIZ. (**D**) The TAD and catalytic domain deletion mutants of GLP are generated. (**E**) The TAD domain of GLP is important for the interaction with WIZ. (**F**) Lacking the catalytic domain of GLP abolishes the interaction with ZNF644. (**G**) The N-terminus deletion mutant of ZNF644 abolishes the interaction with G9a. Myc-tagged ZNF644 and deletion mutants were co-expressed together with SFB-G9a in 293T cells. IP and Western blotting were performed with indicated antibodies. (**H**) The C-terminus deletion mutant of WIZ abolishes the interaction of WIZ and GLP. SFB-tagged WIZ and deletion mutants were co-expressed with Myc-GLP in 293T cells. (**I**) A model shows that the

*Figure 2. continued on next page*

*Figure 2. Continued*
N-terminus of ZNF644 interacts with the TAD of G9a, while the C-terminus of WIZ interacts with the TAD of GLP.
The following figure supplement is available for figure 2:

**Figure supplement 1**. Down-regulation of G9a doesn't affect the interaction between WIZ and GLP.

still tightly associated with genomic DNA (*Figure 3A*). Thus, these results indicate that both ZNF644 and WIZ tightly bind to chromatin. It is also consistent with our affinity purification results that WIZ is slightly difficult to be identified in the mass spectrometry analyses.

Since ZNF644 and WIZ are subunits in the G9a/GLP complex, we ask if ZNF644 and WIZ target the G9a/GLP complex to chromatin. We used siRNAs to knock down ZNF644 and/or WIZ. Lacking either ZNF644 or WIZ impaired the chromatin association of G9a and GLP (*Figure 3B*). Moreover, when we knocked down ZNF644 and WIZ simultaneously, chromatin-bound G9a and GLP were remarkably reduced, but the levels of soluble G9a and GLP were not affected (*Figure 3B*). Reconstituted cells with siRNA-resistant ZNF644 or WIZ retained G9a in the chromatin fraction. However, either the D1 mutant of ZNF644 or the D8 mutant of WIZ was able to target G9a to chromatin (*Figure 3C*). Collectively, these results suggest that ZNF644 and WIZ facilitate the chromatin localization of the G9a/GLP complex (*Figure 3D*).

## WIZ and ZNF644 associate with G9a at specific genomic loci

To determine if ZNF644 and WIZ target G9a to specific genomic loci, we performed high-throughput ChIP sequencing (ChIP-seq) to examine the genome-wide localization of G9a, ZNF644, and WIZ in 293T cells. We identified 14,153 G9a enriched regions, 12,777 ZNF644 enriched regions, and 11,853 WIZ enriched regions, respectively. The ChIP-seq results were validated using ChIP-qPCR to examine 30 randomly picked loci that represent a broad range of ChIP-seq fragment counts (*Figure 4—figure supplement 1*). To analyze genome-wide distribution of those enriched regions, the whole genome was partitioned into three regions: intragenic region, promoter region (5 kb upstream or downstream of the TSS), and distal intergenic region not encoding any genes (*Figure 4—figure supplement 2*). Approximately, 40% of G9a peaks, 45% of ZNF644 peaks, and 43% of WIZ peaks were distributed in gene promoter region (*Figure 4A*). We found around 54% of WIZ-enriched regions were bound by G9a, and around 58% of ZNF644 enriched regions were bound by G9a, while around 63% of G9a enriched regions were bound by ZNF644 and/or WIZ (*Figure 4B*). These results indicate that most G9a-enriched regions are associated with ZNF644 and/or WIZ, which is in agreement with our results that the chromatin loading of G9a is dependent on the ZNF644 and/or WIZ. It has been shown that G9a regulates gene transcription via catalyzing H3K9me2 at promoter regions (*Su et al., 2004*; *Barski et al., 2007*; *Kubicek et al., 2007*; *Chen et al., 2012*; *Fang et al., 2012*). Thus, we analyzed G9a peaks in promoter regions and found that around 82% of G9a-enriched peaks in promoter region were bound by ZNF644 and/or WIZ (*Figure 4C*). Further analyses across G9a peaks in promoter regions show that ZNF644 and WIZ profiles are also associated with the G9a profiles in promoter region (*Figure 4D*). Thus, accumulated evidence suggests that G9a is clearly associated with ZNF644 and WIZ, especially in promoter regions. To further analyze the co-localization of ZNF644 and WIZ with G9a at specific gene loci, we studied several genes with promoter enrichment of G9a. At *CWH43*, *DIP2C* and *ROCK1* loci, G9a, ZNF644 and WIZ co-localized together at the promoter regions (*Figure 4E*). At *CACNA2D1*, *ANKRD26P1*, *USP14,* and *HCN1* loci, only ZNF644, but little WIZ, significantly co-localized with G9a in the promoter regions (*Figure 4F*, *Figure 4—figure supplement 3A*). In contrast, at *PARD3*, *ABCA13*, *SENP5*, and *NRXN3* loci, WIZ, but little ZNF644, co-localized with G9a (*Figure 4G*, *Figure 4—figure supplement 3B*). Taken together, ZNF644 and/or WIZ associate with G9 at the promoter regions of specific loci.

Since ZNF644 and WIZ have 8 and 12 zinc finger motifs, respectively, we further analyzed the DNA-binding sequences of ZNF644 and WIZ based on the information from ChIP-seq. Using Peak-motifs software (http://floresta.eead.csic.es/rsat/peak-motifs_form.cgi), we examined only ZNF644-enriched regions (lacking WIZ), by which we excluded the possible loading of ZNF644 onto chromatin via WIZ.

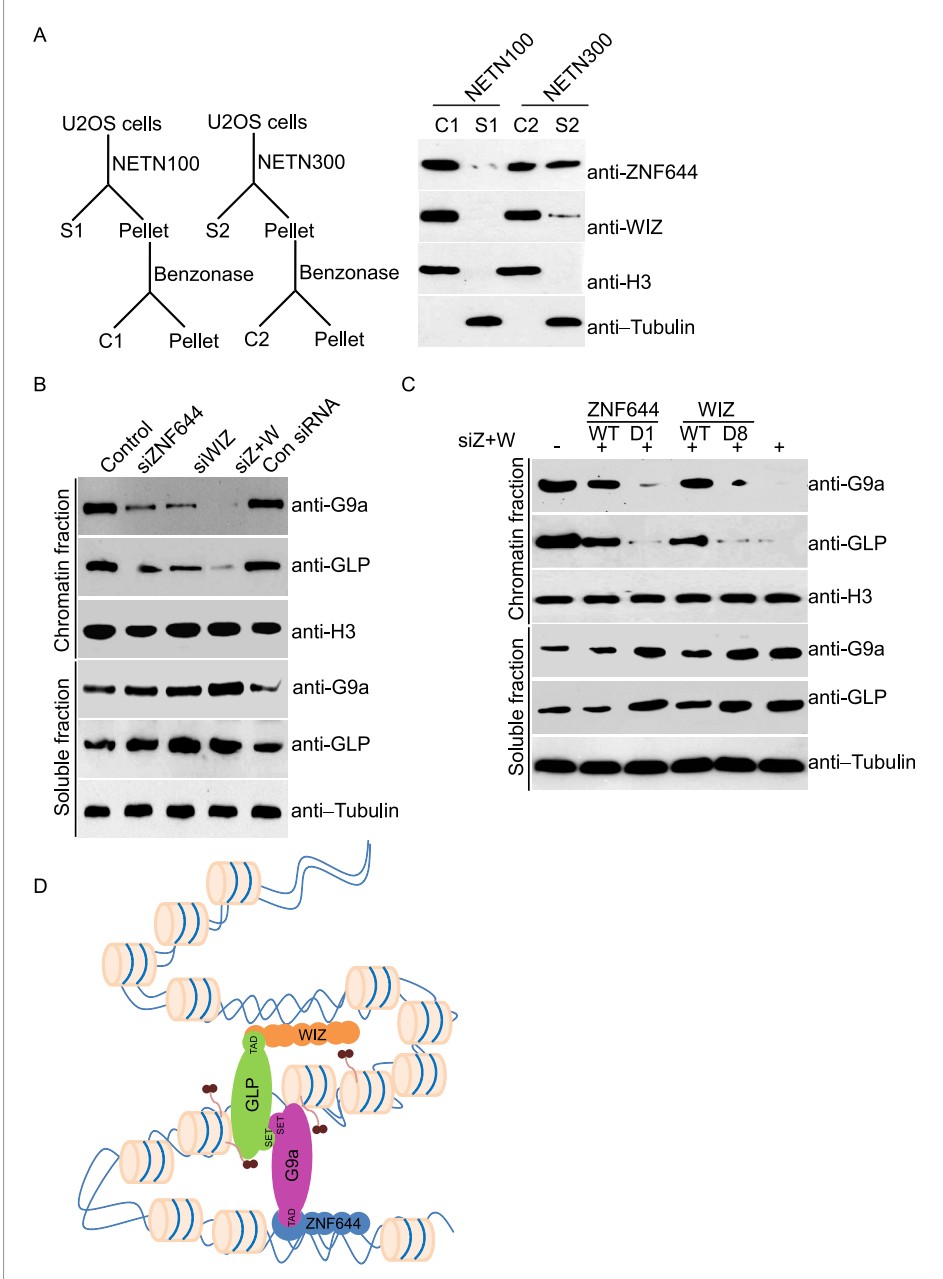

**Figure 3**. ZNF644 and WIZ are important for the chromatin localization of G9a. (**A**) Both ZNF644 and WIZ tightly bind to chromatin. U2OS cells were lysed by NETN100 (lysis buffer with 100 mM NaCl) and NETN300 (lysis buffer with 300 mM NaCl) respectively. After harvesting the soluble fractions, the pellets were digested by Benzonase to extract the chromatin fraction. Each fraction was examined by Western blotting. Tubulin and histone H3 were used as loading control for the soluble fraction and chromatin fraction respectively. (**B**) Knockdown of ZNF644 or/WIZ impairs the chromatin association of G9a and GLP. U2OS cells were lysed with NETN100 buffer. The soluble fraction and chromatin fraction were separated and each fraction was examined with Western blotting. Tubulin and histone H3 were used as loading control in the soluble fraction and chromatin fraction respectively. (**C**) In the cells with siRNA-resistant ZNF644 or WIZ, G9a is retained in the chromatin fraction. But the D1 mutant of ZNF644 or the D8 mutant of WIZ is unable to target G9a to chromatin. (**D**) A model shows that ZNF644 and WIZ facilitate the chromatin localization of the G9a/GLP complex.

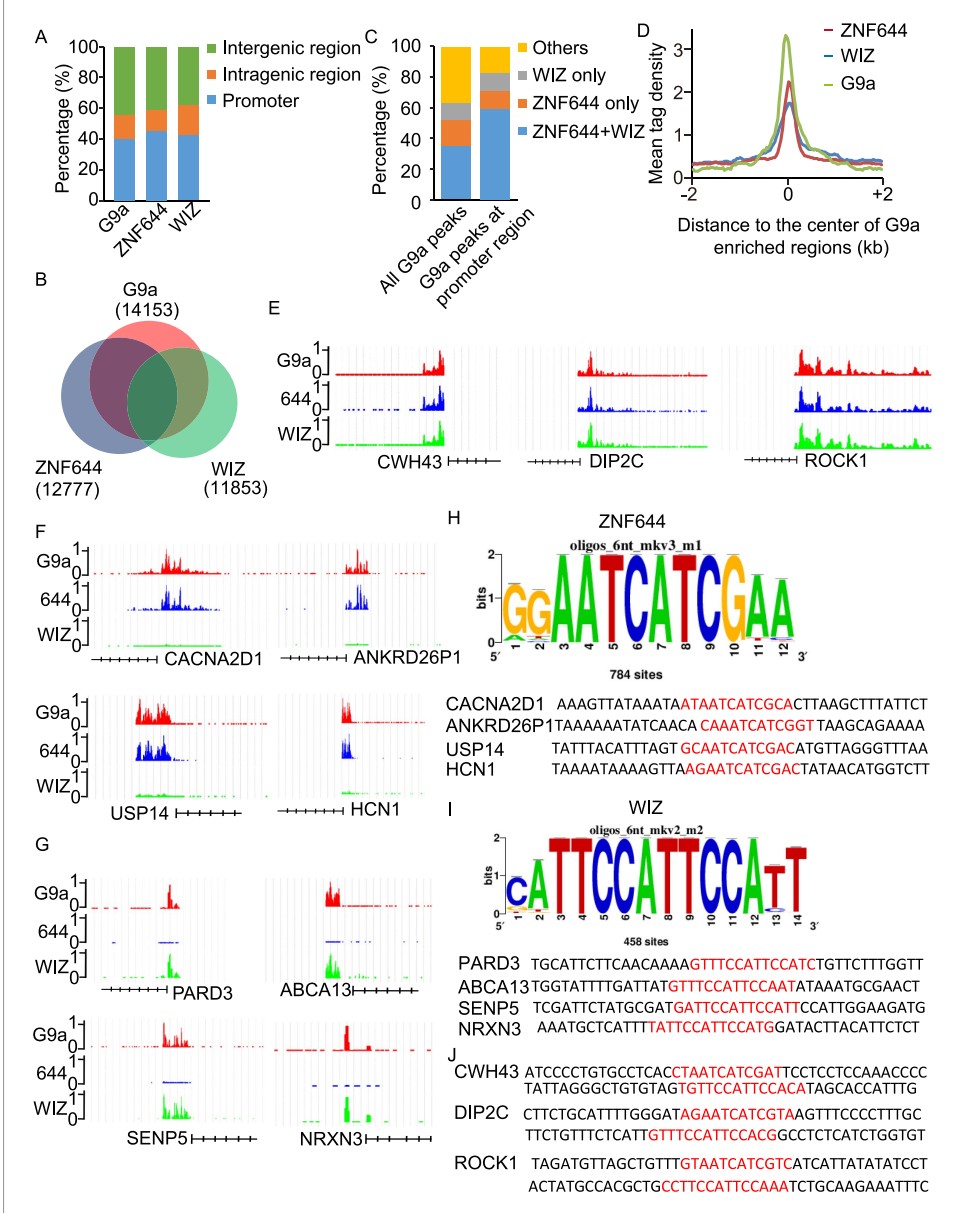

**Figure 4**. WIZ and ZNF644 associate with G9a at specific genomic loci. (**A**) Summary of genome-wide distribution of G9a, ZNF644 and WIZ in different regions. Y-axes: percentage of each region in the genome. (**B**) Venn diagram shows a significant overlap between G9a, ZNF644 and WIZ enriched peaks. (**C**) The G9a-enriched peaks were bound with ZNF644 and/or WIZ, especially in promoter region. (**D**) G9a, ZNF644 and WIZ ChIP-seq read counts in 100-bp window were plotted against the distance (–2 kb, +2 kb) from the center of G9a enriched regions in promoter region. Y-axes: mean tag density. (**E**) ChIP-seq results show the co-occupancy of ZNF644, WIZ and G9a at *CWH43*, *DIP2C* and *ROCK1* loci. (**F**) ZNF644 and G9a are co-localized at the promoter regions of *CACNA2D1*, *ANKRD26P1*, *USP14* and *HCN1*. (**G**) WIZ and G9a are co-localized at the promoter regions of *PARD3*, *ABCA13*, *SENP5* and *NRXN3*. (**H**) The consensus DNA-binding motif of ZNF644 is analyzed according to ChIP-seq result. The binding sequences in *CACNA2D1*, *ANKRD26P1*, *USP14* and *HCN1* loci are shown in red. (**I**) The specific DNA binding sequence of WIZ is obtained according to the ChIP-seq results, and is confirmed at *PARD3*, *ABCA13*, *SENP5* and *NRXN3* loci. (**J**) Both ZNF644 and WIZ-binding sequences are identified at *CWH43*, *DIP2C* and *ROCK1* loci, which are co-occupied by ZNF644 and WIZ.

The following figure supplements are available for figure 4:

*Figure 4. continued on next page*

*Figure 4. Continued*

**Figure supplement 1**. Validation of ChIP-seq results by qPCR.

**Figure supplement 2**. Genome-wide analysis of ChIP-seq peaks.

**Figure supplement 3**. The gene loci occupied by ZNF644 or WIZ are confirmed by ChIP-qPCR.

**Figure supplement 4**. DNA binding motifs of ZNF644 or WIZ concluded from ChIP-seq results were validated by Electrophoretic Mobility Shift Assay (EMSA).

A specific DNA-binding sequence was concluded by the software (*Figure 4H*), and this sequence was confirmed at *CACNA2D1, ANKRD26P1, USP14,* and *HCN1* loci only occupied by ZNF644. Similarly, a specific DNA binding sequence of WIZ was also obtained from software analyses (*Figure 4I*) and was confirmed at *PARD3, ABCA13, SENP5,* and *NRXN3* loci. Moreover, both ZNF644 and WIZ-binding sequences were identified at the loci co-occupied by ZNF644 and WIZ, such as *CWH43, DIP2C,* and *ROCK1* loci (*Figure 4J*). We performed electrophoretic mobility shift assays (EMSA) and found that the consensus DNA-binding motifs of ZNF644 and WIZ showed strong binding with full-length recombinant proteins (*Figure 4—figure supplement 4*).

## ZNF644 and WIZ target G9a for gene repression

Since the G9a/GLP complex catalyzes methylation of H3K9 in euchromatin and represses gene transcription (*Tachibana et al., 2005, 2008*), we next explored the function of ZNF644 and WIZ in G9a-dependent gene transcriptional repression with ChIP assays. In agreement with the ChIP-seq results, ZNF644, WIZ, and G9a localized at *CWH43, DIP2C,* and *ROCK1* loci (*Figure 5—figure supplement 1*). Moreover, down-regulation of ZNF644 and WIZ impaired the localization of G9a at these loci (*Figure 5A*), suggesting that ZNF644 and WIZ are important for targeting G9a to this specific loci. However, down-regulation of G9a did not affect the chromatin localization of ZNF644 and WIZ at these loci (*Figure 5—figure supplement 2*). Since G9a catalyzes H3K9me2 that is recognized by HP1α, we found that both H3K9me2 and HP1α were enriched at these loci. Lacking G9a abolished the enrichment of H3K9me2 and HP1α (*Figure 5B*). Similarly, loss of ZNF644 and WIZ also impaired the enrichment of H3K9me2 and HP1α (*Figure 5B*), suggesting that ZNF644 and WIZ are important for the G9a-dependent H3K9 methylation at G9a targeting genes. Since G9a mainly occupied the promoter regions at these gene loci, G9a-dependent H3K9 methylation is likely to repress gene transcription. Down-regulation of G9a by siRNA indeed increased gene transcription at these loci (*Figure 5C*). Again, loss of ZNF644 and WIZ also facilitated gene transcription (*Figure 5C*). Taken together, these results suggest that ZNF644 and WIZ regulate the function of G9a during transcription. Moreover, we examined only ZNF644 or WIZ occupied gene loci, and similar results were obtained (*Figure 5—figure supplement 3*).

To study if the interaction domains in ZNF644 and WIZ are important for the G9a-dependent function, we reconstituted siRNA-treated cells with siRNA-resistant ZNF644 and WIZ. ZNF644 and WIZ rescued the recruitment of G9a to the gene loci, facilitated the enrichment of H3K9me2 and HP1α, and repressed gene transcription (*Figure 6*). However, expression of the D1 mutant of ZNF644 and the D8 mutant of WIZ that abolish the interactions with G9 and GLP, failed to restore the enrichment of H3K9me2 and HP1α as well as transcription repression (*Figure 6*). Thus, our results demonstrate that ZNF644 and WIZ are two key subunits in the G9/GLP complex to target G9a and GLP to genomic loci for transcriptional repression.

## Discussion

In this study, with unbiased protein affinity purification, we identified ZNF644 and WIZ as two core subunits in the G9a/GLP complex. With the analyses on the internal deletion mutants, we found that the N-terminus of ZNF644 interacts with the TAD of G9a, while the C-terminus of WIZ interacts with the TAD of GLP. In the previous study (*Ueda et al., 2006*), WIZ was found to interact with the catalytic

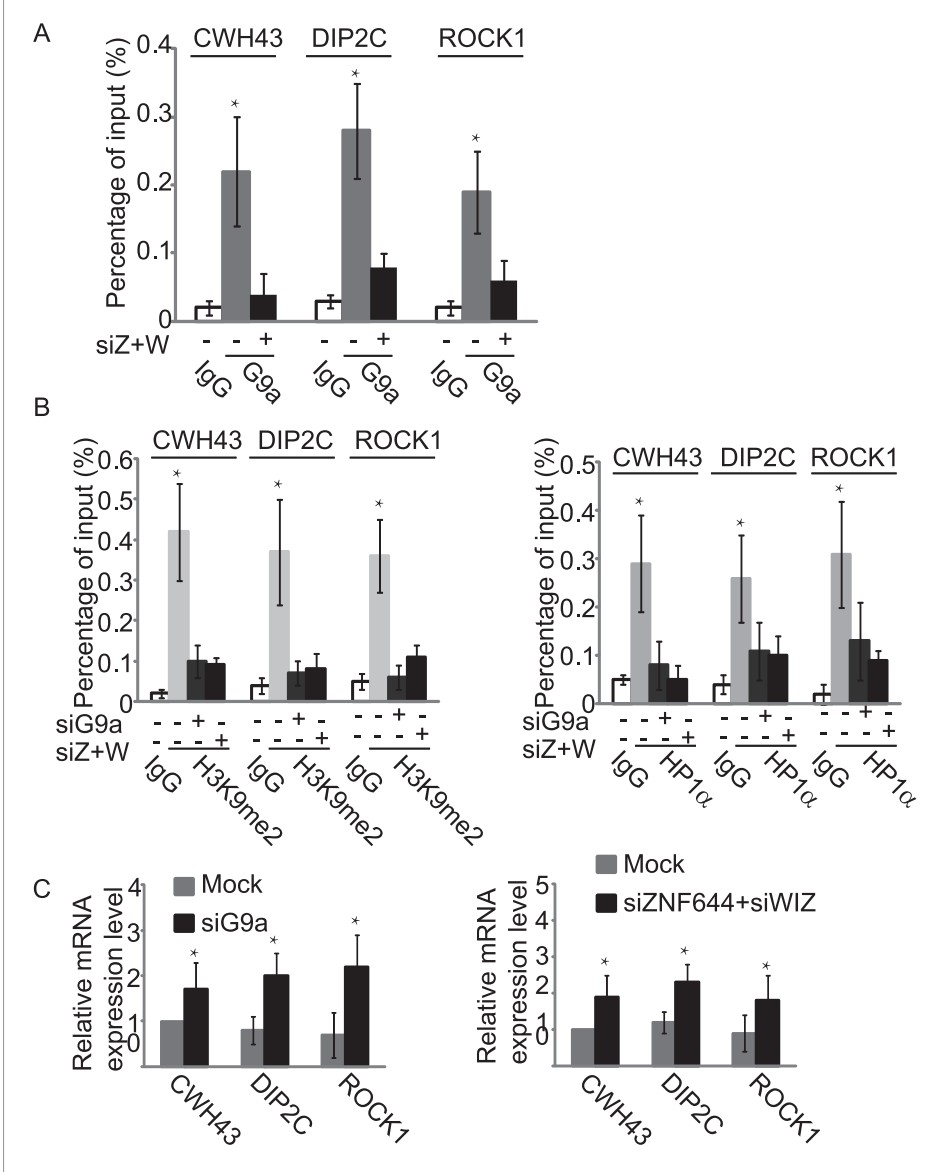

**Figure 5**. ZNF644 and WIZ target G9a for gene repression. (**A**) Down-regulation of ZNF644 and WIZ by siRNAs (siZ + W) impairs the localization of G9a at *CWH43*, *DIP2C* and *ROCK1* loci. *p < 0.05 compared to IgG. (**B**) Knockdown G9a abolishes the enrichment of H3K9me2 and HP1α at *CWH43*, *DIP2C* and *ROCK1* loci. Loss of ZNF644 and WIZ also impairs the enrichment of H3K9me2 and HP1α at these loci. *p < 0.05 compared to IgG. (**C**) Down-regulation of G9a by siRNA increases gene transcription at *CWH43*, *DIP2C* and *ROCK1* loci, and loss of ZNF644 and WIZ also facilitates gene transcription at these loci. *p < 0.05 compared to Mock.

The following figure supplements are available for figure 5:

**Figure supplement 1**. Co-occupancy of ZNF644, WIZ and G9a is shown at *CWH43*, *DIP2C* and *ROCK1* loci.

**Figure supplement 2**. Down-regulation of G9a does not affect the chromatin localization of ZNF644 (**A**) and WIZ (**B**) at *CWH43*, *DIP2C* and *ROCK1* loci.

**Figure supplement 3**. ZNF644 and WIZ target G9a for gene repression at only ZNF644 or WIZ occupied gene loci.

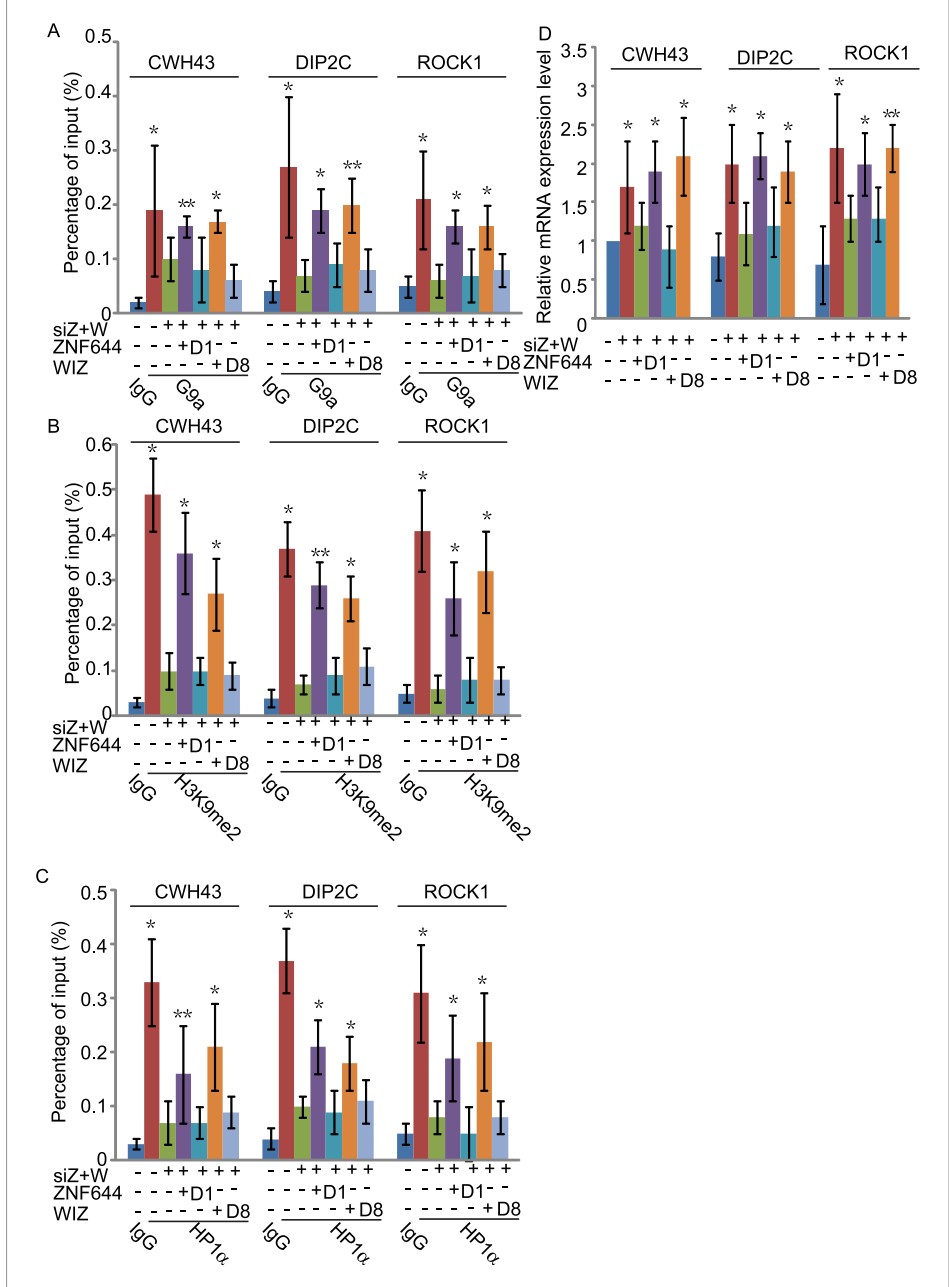

**Figure 6**. The interaction domains in ZNF644 and WIZ are important for the G9a-dependent function. (**A**) Wild-type ZNF644 and WIZ, but not the D1 mutant of ZNF644 and the D8 mutant of WIZ, rescue the recruitment of G9a to *CWH43*, *DIP2C* and *ROCK1* loci. *p < 0.05, **p < 0.01 compared to IgG. (B-D) Wild-type ZNF644 and WIZ, but not the D1 mutant of ZNF644 and the D8 mutant of WIZ, restore the enrichment of H3K9me2 and HP1α as well as gene transcription at *CWH43*, *DIP2C* and *ROCK1* loci. *p < 0.05, **p < 0.01 compared to IgG (**B**, **C**) or the control U2OS cells without siRNAs treatment (**D**).

domain of G9a. We obtained the similar result (*Figure 2C*). However, the catalytic domain of G9a also interacts with the catalytic domain of GLP to form a heterodimer. Lacking G9a did not impair the interaction between WIZ and GLP (*Figure 2—figure supplement 1B*), suggesting that the interaction between WIZ and G9a might be indirect and mediated by the catalytic domain heterodimer of G9a and GLP.

The G9a/GLP complex is known to associate with specific gene promoter and catalyze local H3K9 methylation for transcription repression (*Tachibana et al., 2005*, *2008*). However, both G9a and GLP lack the DNA recognition domains for targeting specific genes. Here, we show the evidence that ZNF644 and WIZ, two multi zinc finger motif-containing proteins, mediate the recruitment of G9a and GLP to the specific gene loci. ZNF644 has 8 zinc finger motifs, while WIZ contains 12 zinc finger motifs. It is likely that these zinc finger motifs function together to recognize specific DNA sequences. Based on our ChIP-seq analyses, we identified the DNA sequences recognized by ZNF644 and WIZ. Thus, ZNF644 and WIZ act as two hands to grab the genomic DNA and target enzymatic subunits G9 and GLP for H3K9 methylation. It is also interesting to notice that ZNF644 and WIZ bind the TADs of G9a and GLP, respectively. Thus, the complex may form a symmetric structure with ZNF644 and G9a on one side, and WIZ and GLP on the other side. These two parts are linked by the interaction between the catalytic domains of G9a and GLP in the middle (*Figures 2I and 3D*). The double DNA recognition may reduce the flexibility of the complex on the chromatin and allow the G9a and GLP to precisely catalyze histone methylation at gene loci.

In the ChIP-seq analysis, we also notice that a fraction of G9a and GLP only associate with either ZNF644 or WIZ (*Figure 4A,B*). Thus, it is possible that ZNF644 or WIZ alone is sufficient for targeting G9a and GLP to certain loci for H3K9 methylation and transcription repression. Since ZNF644 and WIZ bind different DNA sequences, in these cases, different DNA-binding subunits target G9a and GLP to different loci. With only one DNA-binding arm, the G9a/GLP complex may have more flexibility to methylate targets. Alternatively, we cannot rule out the possibility that a small amount of G9a or GLP only form homodimers. Since ZNF644 only recognizes the TAD of G9a, and WIZ interacts with the TAD of GLP, only ZNF644 or WIZ is sufficient to target the homodimer to the chromatin. Moreover, a small set of G9a associates with neither ZNF644 nor WIZ. Thus, it is possible that a small set of G9a may interact with other regulators. It has been shown that G9 associates with other zinc finger proteins, such as Blimp-1, which may also target G9a to the substrates (*Gyory et al., 2004*). However, Blimp-1 is mainly expressed in plasma cells as the major function of Blimp-1 is to regulate plasma cell differentiation (*Shaffer et al., 2002*). Thus, Blimp-1 mainly regulates G9a's activity in plasma cells.

It has been shown that G9a regulates gene transcription via catalyzing H3K9me2 at promoter regions (*Su et al., 2004*; *Barski et al., 2007*; *Kubicek et al., 2007*; *Chen et al., 2012*; *Fang et al., 2012*). Consistently, we found that G9a associated with ZNF644 and WIZ, especially in the promoter regions, to regulate transcription. Interestingly, *Wen et al. 2009* examined H3K9me2-enriched loci in the differentiated tissues and found that large chromatin regions associate with H3K9me2. These regions were named as large organized chromatin K9 modifications (LOCKs). However, the function of LOCKs remains unclear. Interestingly, LOCKs are dynamically regulated during development, and the size of LOCKs varies in different types of cells during differentiation, suggesting that LOCKs might be regulated by not only histone methyltransferases but also demethylases. It is possible that the G9a complex plays a key role for LOCKs formation. However, LOCKs do not exist in cancer cells (*Wen et al., 2009*). Future analysis of the G9a complex during tissue development and differentiation may reveal the mechanism and function of LOCKs. It is possible that, besides ZNF644 and WIZ, other functional partners of G9a regulate LOCKs.

Nevertheless, in this study, we have demonstrated that ZNF644 and WIZ are two major functional partners of G9a and GLP. ZNF644 and WIZ target the G9a/GLP complex to genomic loci for H3K9 methylation and transcription repression.

## Materials and methods

### Plasmids, antibodies, and siRNAs

Full-length cDNA of G9a, GLP, and WIZ was cloned into pS-FLAG-SBP (SFB) vector, respectively, the full-length cDNA of ZNF644, G9a, and GLP was cloned into pCMV-Myc vector, and the full-length cDNA of WIZ was also cloned into pCMV-HA vector. For protein co-immunoprecipitation experiments, G9a deletion mutants, WIZ deletion mutants, and GLP deletion mutants were cloned into SFB vector, respectively. ZNF644 deletion mutants were cloned into the pCMV-Myc vector.

Primary antibodies used in this study include: mouse anti-G9a monoclonal antibody (Abcam, Cambridge, UK), mouse anti-HA and anti-Myc monoclonal antibodies (Covance, Princeton, NJ), rabbit anti-H3K9me2 polyclonal antibody (Upstate, Billerica, MA), rabbit anti-human ZNF644 antibody

(raised against N-terminus a.a. 50–602), rabbit anti-human WIZ antibody (raised against N-terminus a.a. 220–750).

The siRNAs targeting G9a, GLP, ZNF644, and WIZ were ordered from Dharmacon (Lafayette, CO).

## Protein purification

Purification of SFB triple-tagged protein was described previously (*Zhang et al., 2009*). To search for binding partners of G9a or ZNF644, we harvested 50 10 cm$^2$ plates of 293T cells stably expressing SFB-G9a or ZNF644 and washed cells with PBS. Cells were lysed with 30 ml ice-cold NETN300 buffer (0.5% NP-40, 50 mM Tris–HCl pH 8.0, 2 mM EDTA, and 300 mM NaCl). The soluble fraction was incubated with 0.5 ml streptavidin-conjugated agarose beads. The beads were washed with NETN buffer three times. Associated proteins were eluted with 2 mM biotin in PBS and further incubated with 50-ml S beads (Novagen, Billerica, MA). The bound proteins were eluted with SDS sample and analyzed with 10% SDS-PAGE and mass spectrometry.

## Cell lysis, immunoprecipitation, streptavidin beads pull-down, and Western blotting

For immunoprecipitation assays, 293T cells or U2OS cells were lysed with ice-cold NETN400 buffer (0.5% NP-40, 50 mM Tris–HCl pH 8.0, 2 mM EDTA, and 400 mM NaCl) containing 10 mM NaF and 50 mM β-glycerophosphate. The soluble fractions were collected and diluted to 100 mM NaCl, then directly subjected to electrophoresis or immunoprecipitation with indicated antibodies followed by Western blotting analysis with indicated antibodies. For the SFB-tagged protein, streptavidin beads were used to perform the pull-down assay followed by Western blotting analysis.

## Chromatin immunoprecipitation assay

Chromatin immunoprecipitation assays (ChIP) were performed according to the protocol described by Upstate (Billerica, MA). The genomic DNA isolated from 293T cells was sonicated to an average size between 300 and 600 bp. Solubilized chromatin was immunoprecipitated with the antibody against WIZ, G9a, or ZNF644. Antibody–chromatin complexes were pulled-down using protein A-sepharose, washed, and then eluted. After cross-link reversal and proteinase K treatment, immunoprecipitated DNA was extracted with phenol-chloroform, ethanol precipitated, treated with RNase, and dissolved with TE buffer. ChIP DNA was qualified using PicoGreen.

## ChIP sequencing

DNA fragments isolated from ChIP were repaired to blunt ends by T4 DNA polymerase and phosphorylated with T4 polynucleotide kinase using the END-IT kit (Epicentre, Madison, WI). A single 'A' base was added to 3′ end with Klenow. Double-stranded adaptors (75 bp with a 'T' overhang) were ligated to the fragments with DNA ligase. Ligation products between 200 and 600 bp were gel purified to remove unligated adaptors and subjected to 20 PCR cycles. Completed libraries were quantified with PicoGreen. The DNA libraries were analyzed by Solexa/Illumina high-throughput sequencing. The read quality of each sample was determined by FastQC software. After prefiltering the raw data by removing sequence adaptors and low quality reads, the tags were mapped to the human genome (hg19) by Bowtie software. Parameters settings were listed as follows: -v, 3 (reported alignments with at most 3 mismatches), -5, 3 and -3, 7 (trim 3 bases from 5′ end and 7 from 3′ end to remove low-quality bases). Peak detection was performed using MACS software from Galaxy browser (http://galaxyproject.org/). Parameters settings were as follows: IgG ChIP-seq aligned reads were used as control file, tag size with 25 bp, band width with 300 bp. When comparing peaks from different samples, peaks were considered to be overlapping if they were within 2 kb of each other. The peaks obtained from ChIP-seq were matched to the annotated reference genome (human hg19) using Cisgenome 2.0. To view the peak density and position, Cisgenome 2.0 was used. To obtain the binding motif of ZNF644 and WIZ, the online software Peak-motifs http://floresta.eead.csic.es/rsat/peak-motifs_form.cgi) was used. A set of 30 PCR primer pairs (*Supplementary file 1*) were designed to amplify ~200 bp fragments from genomic regions showing a wide range of signals for G9a, ZNF644, and WIZ. ChIP-qPCR values reflect two independent ChIP assays, and each was evaluated in duplicate by qPCR.

To examine the genome distribution of G9a, ZNF644, and WIZ, the whole genome was partitioned into three regions: intragenic region, promoter region (5 kb upstream or downstream of the TSS), and distal intergenic region that does not encode any genes. Genes not uniquely mapped to the genome were excluded. To avoid redundancy, only the longest transcript variant of each gene was used to define chromosomal locations of the intragenic region, promoter region, and intergenic region.

The read counts around the center of G9a-enriched peaks in promoter region were analyzed by SEQMINER software (Ye et al., 2011). The center of G9a-enriched peaks in promoter region was used as the reference. Tag densities from each ChIP-seq were collected within a window of 4 kb around reference coordinates. The tag density of each ChIP-seq in a 200 bp window was calculated and plotted against distance from the center.

For *Figure 4—figure supplement 2*, genes were profiled 5 kb upstream of the transcriptional start site (TSS), through the gene body and 5 kb downstream of the transcriptional end site (TES). 5 kb upstream of the TSS and 5 kb downstream from the TES were divided into windows of 200 bp, and read counts were calculated in each window. For gene body plots, each gene was segmented into 300 non overlapping windows. Plots were made using a 1 kb moving average. Values are tag-normalized and reflect the number of tags observed in each window.

ChIP-seq data have been deposited in the Gene Expression Omnibus under accession number GSE62616.

## Recombinant proteins

Recombinant proteins were purified from Sf9 insect cells. For generating baculovirus, DNA fragments containing full-length human ZNF644, N-terminus of ZNF644 (a.a. 1–300) (ZNF644N300), full-length human WIZ, and N-terminus of WIZ (a.a. 1–200) (WIZN200) were subcloned into pFastBac Vector with a GST tag. Baculoviruses were generated in accordance with the manufacturer's instructions (Invitrogen, Carlsbad, CA). After Sf9 cells were infected with baculoviruses for 48 hr, the cells were harvested, washed with cold PBS three times and lysed with ice-cold NETN100 buffer (20 mM Tris–HCl pH 8.0, 100 mM NaCl, 1 mM EDTA, 0.5% Nonidet P-40). The soluble fraction was incubated with Glutathione–Sepharose beads and eluted with Glutathione.

## Electrophoretic mobility shift assay (EMSA)

Oligonucleotide substrates were obtained from IDT (IDT, Coralville, IA) and were purified by polyacrylamide gel electrophoresis (PAGE). The following oligonucleotides containing ZNF644 binding motif and WIZ binding motif were used, 5′- GAGTAAGATCATGCCACTG***GGAATCATCGAA***CACAGAGTGAGGC TGGG -3′ (ZNF644 'WT' DNA target);

5′- GAGTCTCACTCACGCGC***CATTCCATTCCATT***CAGATACTAGTACGGTCAG -3′ (WIZ 'WT DNA target'). The oligonucleotides containing ZNF644 binding motif mutation and WIZ binding motif mutation were used as control, 5′- GAGTAAGATCATGCCACTG***GCATTGTTGACT***CACAGAGTGAGGCTGGG -3′ (ZNF644 'mutant' DNA target);

5′- GAGTCTCACTCACGCGC***TGCAATCAGGAA***CAGATACTAGTACGGTCAG -3′ (WIZ 'mutant' DNA target). 48-mer oligonucleotides were annealed at 1:1 molar ratio to its complementary oligonucleotides to generate the dsDNA and then radio-labeled with $^{32}$P at the 5′-end. GST-ZNF644, GST-ZNF644N300, GST-WIZ, or GST-WIZN200 was incubated with 0.2 nM (molecules) radio-labeled DNA substrates for 2 hr at 4°C in buffer D (20 mM HEPES-KOH (pH 7.9), 20% glycerol (vol/vol), 0.2 mM EDTA, 0.1 M KCl, 0.5 mM PSMF, 1 mM DTT) with 1.25 µg/µl Bovine serum albumin, 1 mM DTT, 5 mM MgCl$_2$. The samples were resolved by electrophoresis on a 7.5% polyacrylamide gel in TBE buffer for 70 min at 60 V. The gel was then dried and exposed to autoradiography film overnight.

## Statistical analysis

In all cases, multiple independent experiments were performed on different days to verify the reproducibility of experimental findings. Two-way comparison was performed using the *t*-test, and ANOVA was used for more than two groups. For all analyses, a p value of less than 0.05 was considered significant. Results are given as means ± s.d.

## Acknowledgements

This work was supported by grants from National Institutes of Health (CA132755, CA130899, and CA187209 to XY). XY is a recipient of Era of Hope Scholar Award from the Department of Defense. We thank Sean Lamarche for proofreading.

## Additional information

### Funding

| Funder | Grant reference | Author |
|---|---|---|
| National Institutes of Health (NIH) | CA132755 | Xiaochun Yu |
| National Institutes of Health (NIH) | CA130899 | Xiaochun Yu |
| National Institutes of Health (NIH) | CA187209 | Xiaochun Yu |
| U.S. Department of Defense | Era of Hope Scholar Award | Xiaochun Yu |

The funders had no role in study design, data collection and interpretation, or the decision to submit the work for publication.

### Author contributions

CB, Acquisition of data, Analysis and interpretation of data, Drafting or revising the article, Contributed unpublished essential data or reagents; QC, Acquisition of data, Analysis and interpretation of data; XY, Conception and design, Analysis and interpretation of data, Drafting or revising the article, Contributed unpublished essential data or reagents

## Additional files

### Supplementary file

• Supplementary file 1. PCR primers. All the primers used for RT-PCR and ChIP-qPCR are listed.

### Major dataset

The following dataset was generated:

| Author(s) | Year | Dataset title | Dataset ID and/or URL | Database, license, and accessibility information |
|---|---|---|---|---|
| Bian C, Yu X | 2014 | G9a, ZNF644 and WIZ ChIP-seq results | http://www.ncbi.nlm.nih.gov/geo/query/acc.cgi?acc=GSE62616 | Publicly available at NCBI Gene Expression Omnibus (GSE62616). |

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
