## [Decision Letter]

Thank you for sending your manuscript titled “ZNF644 and WIZ regulate the G9a/GLP complex for gene repression” for consideration for publication in *eLife*. Your manuscript was reviewed by three experts in the field and by a member of the Board of Reviewing Editors (BRE). After a full discussion of the study and the reviews, we are happy to report that the reviewers and the BRE member found the study of interest to the journal and therefore, we will be pleased to consider a revised manuscript addressing the following issues:

1) Provide statistical analyses and significance for the quantitative PCR studies and for the data on changes in gene expression when the G9a-GLP complex components are depleted.

2) Provide comprehensive genome-wide analysis including metagene, whole gene distributions, and average promoter analysis for the observations made in the manuscript.

3) Provide experimental evidence for direct interactions of ZNF644 and WIZ to the DNA sequence studied in the manuscript.

4) Provide a deeper discussion between the current analysis and the previous studies in terms of domains of interaction between WIZ and G9a-GLP. Also clarification on the difference between Feinberg's studies on the requirement of megabase size domains of histone H3K9me2 Vs the current study on the role of promoters in this process.

5) The manuscript needs to be read carefully for grammar and clarity.

Further details can be found in the comments below.

Reviewer #1

Significance: It was previously known that WIZ associates with and stabilizes the G9a-GLP heterodimer, but this paper identifies a new binding partner ZNF644. Furthermore, in a major breakthrough the paper demonstrates that WIZ and ZNF644 bind directly to specific DNA sequences and thereby target G9a to specific chromosomal locations. Previously, it was known that a few repressive DNA binding transcription factors can recruit G9a to specific chromosomal loci, but this paper describes a major new mechanism for targeting G9a and GLP to chromatin. The scientific quality of the data presented is generally quite good, but with a few exceptions noted below. There are significant English problems with the manuscript.

Major concerns: The quantitative PCR data for ChIP and for RNA expression throughout Figures 4, 5 and 6 (including the supplements) lack any statistical analysis. The meaning of the error bars is not indicated as well as whether they represent technical or biological replicates and how many. Reproducibility in multiple biological replicates is not discussed. In many cases in these figures the error bars suggest that the differences seen and described in the text may not be significant. In particular, the changes in target gene RNA levels caused by depleting components of the G9a-GLP complex are quite small and appear to be of questionable statistical significance.

Overall recommendation: This is an exciting paper because it reports a new mechanism for targeting of G9a-GLP to specific chromosomal locations. Statistical analyses need to be provided for the quantitative PCR data. Furthermore, the data on changes in gene expression when G9a-GLP complex components are depleted is weak and of questionable statistical significance. This data should either be improved (at least to statistical significance) or eliminated. While it would be nice to have an indication that G9a-GLP is having an effect on gene expression at the sites where they are recruited, the novel targeting mechanism would still, in my opinion, be important enough to merit publication. English needs to be improved throughout the manuscript.

Reviewer #2

G9a-GLP is important for assembly of H3K9 dimethylated facultative heterochromatin. A major question is how does the complex localize to its sites of action? This is a key question in the field of chromatin biology. The authors show that the zinc finger TFs WIZ and ZN644 associate with G9a-GLP in vivo. They map interacting domains by co-IP with deletion mutants and present genome wide ChIP data on co-localization of the complex and locus-specific ChIP in various RNAi mutants. The interaction with WIZ was previously shown so the novelty here is the association with ZN644, the identification of binding motifs and validation of co-localization genome wide. These are important findings of broad interest in the field.

The major drawbacks are the lack of a comprehensive genome wide analysis (i.e., metagene, whole gene distributions, average promoter analysis and so on). Additionally, while the authors show strong evidence for targeting of G9a-GLP by WIZ and ZNF644, this is not linked globally with H3K9 dimethylation. There is an overemphasis on locus-specific ChIP studies that lacked measures of statistical significance. Moreover, the paper lacked a deeper discussion of and comparison with other discoveries in the field. For example, there is some disagreement between the authors' analysis and previous studies in terms of domains of interaction between WIZ and G9a-GLP. Additionally, Feinberg showed megabase size domains of histone H3K9me2, while the authors' studies focus on the promoters. Is WIZ-ZNF644-G9a-GLP binding inside as well as at gene promoters? Finally, there are a huge number of disappointing grammatical issues.

In general, I felt that the paper was a bit underdeveloped throughout but has the potential to make a significant contribution.

Reviewer #3

The manuscript from Bian and colleagues describes the interaction of G9A/GLP heterodimeric methyltransferase complex with two zinc finger containing proteins, Wiz and ZNF644. G9A/GLP activity results in heterochromatin formation and gene silencing. Affinity purification of G9A and ZNF644 was used to determine that G9A/GLP, Wiz and ZNF644 form a core complex. Wiz was shown previously to associate with G9A/GLP and alter its stability (Udea et al.); however, ZNF644 is a new component identified by this manuscript. Using immunoprecipitations of deletion mutants the authors determine that the TAD Domain of GLP and G9A interact selectively with WIZ and ZNF644, respectively. The authors go on to identify the genomic location of WIZ and ZNF644 and that the sites occupied by these proteins accounts for the majority of G9A occupied sites in the genome. Recruitment of G9A and the downstream readers of H3K9 methylation are effected by suppressing WIZ expression. Overall, the data make a compelling case for an important role of WIZ and ZNF644 in recruiting G9A/GLP to their genomic locations. To fully support the proposed role of these proteins in recognition of select genes through a defined DNA sequence, the following points should be addressed:

1) This implication of the requirement of ZNF644 or WIZ for GLP/G9A recruitment is that the zinc-finger domains of these proteins mediate sequence specific DNA interactions. Although the authors identify motifs within genes that are bound by the GLP/G9A/WIZ/ZNF644 complex, the ability of ZNF644 and WIZ to directly bind these DNA sequences is not addressed. The authors should determine if WIZ or ZNF644 selectively and directly recognize the motifs identified in Figure 4 by gel shift or similar assay.

2) The authors identify deletion mutants in ZNF644 and WIZ that fail to bind GLP and G9A. They go on to use these mutants in Figures 3 and 6 to suggest eliminating the ability of GLP or G9A to bind the zinc finger proteins results in an inability of the methyltransferases to stably associate with DNA and suppress transcription. However, for the data in Figures 3 and 6 to fully support the author's hypothesis the D1 and D8 mutants must be shown to stably interact with the DNA: 1) by selective extraction (Figure 3) and 2) by ChIP. The trivial explanation for the existing data is that these mutants themselves fail to be recruited to DNA.

---

## [Author Response]

*After a full discussion of the study and the reviews, we are happy to report that the reviewers and the BRE member found the study of interest to the journal and therefore, we will be pleased to consider a revised manuscript addressing the following issues*:

*1) Provide statistical analyses and significance for the quantitative PCR studies and for the data on changes in gene expression when the G9a-GLP complex components are depleted*.

Thank you for your suggestion. We have performed the statistical analyses for the quantitative PCR studies and for the data on changes in gene expression, including Figure 4—figure supplement 3, Figure 5, Figure 5—figure supplement 1, Figure 5—figure supplement 2 and Figure 5—figure supplement 3 and Figure 6. Two-way comparison was performed using the t-test, and ANOVA was used for more than two groups. For all analyses, a *P*-value with less than 0.05 was considered as statistically significant difference. Results are given as means ± s.d. The description of statistical analysis was also included in the revised Materials and methods.

*2) Provide comprehensive genome-wide analysis including metagene, whole gene distributions, and average promoter analysis for the observations made in the manuscript*.

As suggested by the reviewers, we re-analyzed the ChIP-seq data. The whole genome was partitioned into three regions: intragenic region, promoter region (5kb upstream or downstream of the TSS), and distal intergenic region that does not encode any gene. As shown in revised Figure 4, a total of 14153 genomic regions are enriched with G9a. Around 40 % of peaks are in promoter region, 16 % are in intergenic region, and 44 % are in intragenic region (revised Figure 4). We found around 63 % of G9a peaks were occupied by ZNF644 and/or WIZ (Revised Figure 4), which is consistent with our biochemistry analyses of the G9a complex. By analyzing the genome-wide distribution of the co-localized regions, we found that around 82 % of G9a-enriched peaks in promoter region were bound by ZNF644 and/or WIZ (revised Figure 4). Further analyses across G9a peaks in promoter regions show that ZNF644 and WIZ profiles are associated with the G9a profiles in promoter region (revised Figure 4). These results were further validated by individual gene analyses (revised Figure 4 E–G). Since both ZNF644 and WIZ are DNA-binding proteins, it suggests that ZNF644 and WIZ may target G9a to promoter regions. The revised text on the comprehensive genome-wide analyses in the Figure 4 was included (subsection headed “WIZ and ZNF644 associate with G9a at specific genomic loci”).

*3) Provide experimental evidence for direct interactions of ZNF644 and WIZ to the DNA sequence studied in the manuscript*.

Following the suggestions of the reviewers, we performed electrophoretic mobility shift assays and examined the interaction between the recombinant ZNF644 or WIZ proteins and oligonucleotides with consensus sequences. Oligos with non-consensus sequence were served as negative control. We also used GST proteins without zinc finger motifs as the pull-down control. ZNF644 and WIZ clearly bind to the specific DNA motif in vitro (Figure 4—figure supplement 4). The results were also included in the revised text (“both ZNF644 and WIZ-binding sequences… with full-length recombinant proteins”).

*4) Provide a deeper discussion between the current analysis and the previous studies in terms of domains of interaction between WIZ and G9a-GLP. Also clarification on the difference between Feinberg's studies on the requirement of megabase size domains of histone H3K9me2 Vs the current study on the role of promoters in this process*.

Thank you for the suggestion. In the previous study (34), WIZ was found to interact with the catalytic domain of G9a. We obtained the similar result (Figure 2). However, the catalytic domain of G9a also interacts with the catalytic domain of GLP to form a heterodimer. Moreover, lacking G9a did not impair the interaction between WIZ and GLP (Figure 2—figure supplement 1), suggesting that the interaction between WIZ and G9a might be indirect and mediated by the catalytic domain heterodimer of G9a and GLP. As suggested, we have included this part into the Discussion section. Future structure analysis will provide the details for the interactions in the G9 complex.

It has been shown that G9a regulates gene transcription via catalyzing H3K9me2 at promoter regions (7; 28; 12; 2; 16). Consistently, we found that G9a associated with ZNF644 and WIZ, especially in the promoter regions, to regulate transcription (revised Figure 4). Wen et al. examined H3K9me2-enriched loci in the differentiated tissues and found that large chromatin regions associate with H3K9me2 (35). These regions were named as large organized chromatin K9 modifications (LOCKs). Interestingly, LOCKs are dynamically regulated during development, and the size of LOCKs varies in different types of cells during tissue development and differentiation, suggesting that LOCKs might be regulated by not only histone methyltransferases but also demethylases. It is unclear whether LOCKs are exclusively dependent on the G9a complex (35). And the function of LOCKs is unclear. Moreover, LOCKs do not exist in human tumor cell lines (35). Nevertheless, LOCKs might be very important chromatin modifications. And we have discussed the possible link between LOCKs and the G9 complex (in the beginning of the Discussion section). It is possible that the G9a complex dynamically regulates LOCKs. However, due to current research setting using 293T cells (human tumor cells), we could not analyze LOCKs in our system. It is possible that both WIZ and ZNF644 regulate LOCKs during development. But we could not rule out other possibilities such as demethylases or other functional partners of G9 specifically expressed during development.

*5) The manuscript needs to be read carefully for grammar and clarity*.

Thank you for the reminder, the revised manuscript has been proofread by a native English speaker in our institution.